# Variation in Infant Formula Macronutrient Ingredients Is Associated with Infant Anthropometrics

**DOI:** 10.3390/nu12113465

**Published:** 2020-11-12

**Authors:** Bridget Young

**Affiliations:** Department of Pediatrics, Allergy and Immunology, University of Rochester School of Medicine and Dentistry, Rochester, NY 14642, USA; Bridget_Young@URMC.Rochester.edu

**Keywords:** infant formula, infant growth, infant nutrition, palm oil, National Health and Nutrition Examination Survey, NHANES

## Abstract

Background: There is wide variation in the macronutrient ingredient base of infant formula. How variation in macronutrient ingredients may impact infant growth remains largely unknown. Methods: The 2015–2016 National Health and Nutrition Examination Survey (NHANES) dataset was utilized, including infant anthropometrics and dietary intake. The protein, fat, and carbohydrate sources of formulas consumed were assembled and considered as potential predictors in multivariable models of infant Z-scores among infants < 6 months, 6–12 months and all infants combined (0–12 months). Results: The following relationships represent ingredient covariates within the final multivariable models of infant Z-scores. Consuming formula with palm oil was associated with higher weight-for-length Z-scores among infants < 6 months, but lower weight-for-age and weight-for-length Z-scores among infants 6–12 months. Consuming soy-protein formulas was associated with lower weight-for-length, head circumference-for-age and abdominal circumference-for-age Z-scores among infants < 6 months. Consuming sucrose-containing formula was associated with higher weight-for-length and abdominal circumference-for-age Z-score among infants 0–12 months. Conclusions: These data provide proof-of-concept that all formulas are not the same. Variation in macronutrient ingredients within the standard formula category is associated with differences in infant anthropometric outcomes. Long-term and mechanistic studies are warranted to pursue these findings; especially for palm oil, soy protein, and sucrose.

## 1. Introduction

In the United States, the composition of the 29 essential nutrients in infant formula is regulated by the Food and Drug Administration, and all sources of macronutrients used in infant formulas are approved for safety and maintenance of “normal” growth over the first year [1]. Variation in sourcing of macronutrients results in a wide variety of infant formula formulations on the market in the US.

Historical research regarding protein source in formula has primarily focused on soy vs. dairy-based formula [2,3,4], and an extensive focus on the degree of protein hydrolysis as it relates to immune-related conditions, such as eczema and type 1 diabetes [5,6,7,8,9,10,11]. Limited work has focused on growth and shown that both partially and fully hydrolyzed protein sources do result in reduced growth rates (compared to intact protein formula), more similar to breastfed infants [12,13,14]. Little to no research has investigated if the proportion of whey and/or casein in dairy formula has long term implications on growth. Whey has a higher gastric emptying rate and is less allergenic than casein [15,16,17,18]. Human milk is predominantly whey-based whereas bovine milk is predominantly casein based. Common infant formulas on the US market represent composition ranging from 0–100% in each category.

Infant formulas in the US provide fat as a blend of vegetable oils. The focus of previous research regarding infant formula fat sources has been on the effect of palm oils on calcium and fat absorption, and bone mineral accrual [19,20,21,22]. The impact of other common fats, such as coconut and soy oils on variation in growth remains unknown.

The caloric carbohydrate source in breast milk is lactose. Often alternative glucose or sucrose-based carbohydrates are utilized in infant formula as a replacement for lactose. Recently, a single study of 23 formula fed infants reported that consuming a glucose-based sugar (corn syrup solids) in formula, results in significant alterations in postprandial measures of metabolism, compared to lactose [23]. The impact of either glucose or sucrose-based carbohydrates on infant growth and development, or long-term health outcomes remains unstudied.

Give the limited research investigating variation in macronutrient composition of infant formulas on the market and infant growth measures, our aims were to: document the variation in macronutrient intake of a nationally representative sample of formula fed infants in the United States and to investigate any relationships between macronutrient ingredients and infant anthropometric measures.

## 2. Materials and Methods

### 2.1. Infant Formula Nutrition Database

Infant formula macronutrient ingredients and relative proportion of ingredients for all name brand formulas on the US market in 2015–2016 were assembled utilizing product websites, and contacting the company when information was unavailable either on the public or medical practitioner websites.

Protein source was classified as % casein and % whey for dairy-based formulas and categorically yes/no for soy-protein based formulas. Protein size was categorized as either: amino acid, fully hydrolyzed, partially hydrolyzed, or intact protein. Caloric carbohydrate source was classified as a percentage of three categories based on the metabolic byproduct of the ingredient used: lactose, glucose, and sucrose. Corn syrup, corn syrup solids, brown rice syrup, starch, glucose syrup, and maltodextrin were all considered “glucose”. Formulas were also classified categorically as containing sucrose or not. Fat source was classified by percentage of fat that was provided by the following categories: soy oils, high oleic sunflower or safflower oils, and coconut or medium chain triglycerides (MCT) oils. Formulas were also classified categorically as containing palm or palm olein oils or not.

### 2.2. NHANES Data

Data from the National Health and Nutrition Examination Survey (NHANES) 2015–2016 were utilized. Dietary data and clinical data (body measures, dietary interview day 1 and 2, early child questionnaire) were downloaded from the public website [24]. The following age and sex-specific Z-scores were generated from the anthropometric measures in the NHANES dataset utilizing the WHO standards and SAS codes provided by the Center for Disease Control and Prevention (CDC) [25]: Weight-for-age (WAZ), length-for-age (LAZ), weight-for-length (WLZ), head circumference-for-age (HCZ), and abdominal circumference-for-age (ACZ).

Infant diet data were averaged when two days of dietary intake were available. Average daily caloric intake from formula, average number of formula feeds and average number of human milk feeds were calculated. Breastfeeding exclusivity was calculated as the proportion of total breast milk and formula feeds that were breast milk (i.e.: 100% = exclusively breastfed and 0% = exclusively formula fed). Breastfeeding exclusivity was also categorized as exclusively breastfed, vs. mixed-fed, vs. exclusively formula fed. Birth weight was categorized as follows: <2500 g = low birth weight (LBW) and >4000 g = large for gestational age (LGA). Calculated and relevant clinical NHANES variables from each dataset were merged by unique identifier. Only data from infants ≤ 12 months at the time of NHANES exam were kept in the dataset. Formula nutrient composition was merged into this trimmed dataset based on formula name.

### 2.3. Multivariable Modeling

Non-normally distributed variables were either log or square-root transformed before analyses. Simple linear regression (univariate) analysis was used to determine if any of the macronutrient ingredient variables were correlated with infant z-scores. Breastfeeding exclusivity category, total daily kcal intake from formula, consuming solid food (or not) and birth weight category were also considered as potential predictors of Z-scores.

Variables that were independently associated with Z-scores at a significance level of *p* ≤ 0.2 were entered as independent variables into a multivariable model of Z-scores. Backward stepwise regression was then utilized to achieve the most parsimonious model that maximized the model’s adjusted R^2^ (which are reported) with the minimum number of covariates. The normality of all models’ residuals was confirmed.

This modeling was performed among infants < 6 months of age as a means of focusing on infants who’s primary source of calories was breast milk and/or formula, and among older infants (6–12 months) and on the population as a whole (all infants 0–12 months).

Head circumference and abdominal circumference was not collected on every infant. Head circumference was only measured on 13 infants between 6–12 months of age. Thus, HCZ was only modeled for infants < 6 months of age.

Results are reported as means ± standard deviations (SD) unless otherwise noted. For covariate variables that were transformed within models, parameter estimates reported herein were back-transformed in order to be clinically interpretable. Analyses were performed in SAS 9.4 and JMP Pro 14 (SAS Institute Inc, Cary, NC, USA)

## 3. Results

### 3.1. Cohort Characteristics

Sample size numbers for the final dataset generation are presented in Figure 1. A total of 395 infants ≤ 12 months of age at the screen had feeding data available for at least one day. Eighteen infants were excluded because no anthropometric measures were recorded. Twenty-five infants were excluded because they were consuming generic-brand infant formula and the NHANES dataset does not record any deeper resolution on formula type in this case, and thus deriving these infants’ formula macronutrient intake was impossible. Five infants were consuming two different formulas (either between day 1 vs. day 2 of dietary intake or within the same day). In these cases, the weighted mean of these formulas’ composition was calculated and used. One infant’s data was excluded because birth weight was missing. One infant was consuming an elemental (amino-acid based) formula. This infant was excluded from modeling. One infant was consuming a specialized formula for acute treatment of diarrhea which contained a unique fat blend of only soy oil and coconut oil [26]. As a single outlier for soy oil consumption, this infant was excluded from modeling when soy oil intake was considered a covariate. The final sample size of infants included were 351; 232 of these were consuming at least some formula.

Cohort characteristics are presented in Table 1.

### 3.2. Infant Intake

Table 2 reports the average infant formula macronutrient intakes for the cohort as a whole.

Among infants who were consuming sucrose (15.4% of infants), their carbohydrate intake consisted of 22.4 ± 7.8% sucrose. No infants were consuming premature infant formula.

Figure 2 presents the percentage of infants who reported consuming solid foods on the day(s) a 24-h dietary recall was performed, separated by infant age in months.

### 3.3. Modeling Z-Scores

A total of 230 infants ≤ 12 months of age were consuming some formula. Parsimonious multivariable models of each Z-score outcome were generated for infants < 6 months, 6–12 months, and all infants combined (0–12 months) and are presented in Table 3.

#### 3.3.1. Infants < 6 Months

A total of 124 infants < 6 months of age were consuming some formula.

In univariate models of WAZ, protein size (intact vs. partially hydrolyzed vs. fully hydrolyzed), consuming palm oil, % fat as coconut oil/MCT, and % fat as soy oil, and birth weight category were independent predictors at *p* < 0.2. In the final model of WAZ, only birth weight category remained in the model.

In univariate models of LAZ birth weight category was the only predictor at *p* < 0.2.

In univariate models of WLZ, protein size (intact vs. partially hydrolyzed vs. fully hydrolyzed), consuming soy protein, consuming palm oil, % fat as coconut oil/MCT, % of fat as soy oil, total daily kcal formula consumed and breastfeeding exclusivity were independent predictors at *p* < 0.2. In the final multivariable model of WLZ, consuming a partially hydrolyzed protein or consuming soy protein were associated with lower WLZ, while a fully hydrolyzed protein, consuming palm oil, and total daily kcal formula consumed were all associated with higher WLZ (Table 3).

In univariate models of HCZ, consuming soy protein, total daily kcal formula consumed, and birth weight category were independent predictors at *p* < 0.2. In the final model of HCZ, consuming soy protein was associated with a lower HCZ.

In univariate models of ACZ, consuming soy protein, consuming palm oil, % fat as coconut oil/MCT, % fat as soy oil, total daily kcal formula consumed and breastfeeding exclusivity were independent predictors at *p* < 0.2. In the final multivariate model of ACZ, consuming a soy protein formula and the % fat as coconut/MCT oil was associated with lower ACZ, whereas exclusive formula feeding (no human milk) was associated with higher ACZ.

#### 3.3.2. Infants 6–12 Months

A total of 106 infants 6–12 months of age were consuming some formula.

In univariate models of WAZ, consuming palm oil, % fat as soy oil, as well as birth weight category and whether the infant was consuming solid foods were all predictors at *p* < 0.2. In the final multivariable model of WAZ, birth weight category was associated with WAZ and consuming palm oil was associated with a lower WAZ.

In univariate models of LAZ, birth weight category was the only predictor at *p* < 0.2.

In univariate models of WLZ, consuming sucrose, consuming palm oil, birth weight category, and whether the infant was consuming solid foods were all predictors at *p* < 0.2. In the final multivariable model of WLZ, birth weight category was associated with WLZ and consuming palm oil was associated with a lower WLZ.

In univariate models of ACZ, % of protein as casein, consuming sucrose, % fat as coconut oil birth weight category, and whether the infant was consuming solids were independent predictors at *p* < 0.2. In the final multivariable model of ACZ, birth weight category was associated with ACZ and consuming sucrose was associated with a higher ACZ.

#### 3.3.3. Infants 0–12 Months

A total of 232 infants ≤ 12 months of age were consuming some formula. Parsimonious multivariable models of each Z-score outcome were generated and are presented in Table 3.

Of all the variables considered in univariate models of WAZ and LAZ, birth weight category was the only predictor at *p* < 0.2.

In univariate models of WLZ, consuming sucrose, % of fat as soy oil and birth weight category were independent predictors at *p* < 0.2. In the final multivariable model of WLZ, birth weight category was a significant predictor. Consuming a formula containing sucrose, and the proportion of fat consisting of soy oil were both associated with a higher WLZ.

In univariate models of ACZ, consuming soy protein, consuming sucrose, breastfeeding exclusivity, and birth weight category were independent predictors at *p* < 0.2. In the final multivariable model of ACZ, birth weight category was a significant predictor. Consuming a soy protein formula was associated with lower ACZ, whereas consuming a formula with sucrose and exclusive formula feeding (no human milk) were both associated with higher ACZ.

## 4. Discussion

These novel data suggest that standard variation in macronutrient ingredient sources in infant formula may be related to infant growth outcomes. Novel associations include the relationship between consuming sucrose and elevated WLZ and ACZ. However, the limitations of the dataset hinder the reach of conclusions. These data do provide proof-of-concept that variation in formula macronutrient source may impact infant outcomes. The relationships presented make a strong argument that mechanistic longitudinal investigations into each of the ingredients is warranted, and that standard infant formulas should not be considered “all the same”.

Among infants < 6 months, consuming formulas with palm oil was associated with a higher WLZ. It is well documented that palm oil decreases intestinal calcium and fatty acid absorption in infants [21,22,27,28]. This results in lower bone mineral accrual in infants fed palm oil containing formulas [20]. This decrease in bone mineral accrual could potentially explain a higher WLZ observed among infants consuming formula containing palm oil if it impacted bone growth. It is noteworthy that consuming palm oil was associated with a lower WAZ and WLZ among infants 6–12 months. The explanation behind the opposite relationship between consuming palm oil with WLZ at <6 months vs. 6–12 months is unknown. It is possible that when older infants start consuming solid foods (often lower in fat than breast milk/infant formula) and decrease their formula intake, the reduced fat absorption caused by consuming palm oil becomes a higher percentage of their total caloric intake, leading to reduced WAZ at this older age. This topic deserves further research, especially among infants who are not meeting weight gain recommendations.

A novel finding in this dataset was that the consumption of formula with sucrose as a carbohydrate correlated with higher ACZ among infants 6–12 months and higher WLZ and ACZ in infants 0–12 months. Sucrose has a higher glycemic index than lactose. Sucrose, a dimer of glucose and fructose, metabolizes to 50% fructose. Thus, in an infant exclusively consuming a formula with 27% carbohydrate as sucrose and 10.4 g carbohydrate/100 mL (a common formulation), 5.6% of the infant’s calories derive from fructose (via sucrose metabolism). Long-term consequences of infant consumption of sucrose is unknown. In adults, high fructose consumption has been associated with increased liver strain and can contribute to development of non-alcoholic fatty liver [29,30] and metabolic syndrome [31]. However, this occurs mostly when excess caloric intake and/or obesity are co-factors [31,32]. In a study of children and adolescents with non-alcoholic fatty liver disease, a dietary intervention that reduced fructose intake from 6.5 to 4.4% of calories was enough to lower systolic blood pressure, percent body fat and markers of liver stress [33]. Healthy infants remain unstudied and thus it remains unknown if these same mechanisms are active in infants, and if 5.6% of calories as fructose (derived from sucrose) is enough to trigger these mechanisms. Abdominal circumference Z-score is a less standard anthropometric assessment and thus clinical relevance is limited. However, the similar relationship detected with WLZ among all infants highlights the need for such future investigations.

In these data, consuming a soy protein formula was associated with a lower WLZ, HCZ, and ACZ in infants < 6 months, as well as a lower ACZ among all infants (0–12 months). Previous work has shown that infants fed soy formula have different body composition trajectories, exhibiting higher total fat free mass than breastfed and dairy-formula fed infants at 3 and 6 months but lower % fat mass than breastfed infants only up to 6 months [3]. Soy formula fed infants also exhibit lower bone mineral content than breastfed infants until roughly 4 months, but then equivalent or higher bone mineral content by 12 months [3,34]. These altered body composition patterns may explain why this study detects lower WLZ, HCZ, and ACZ at <6 months, but not among infants 6–12 months. However, it is important to note that with all these correlations, reverse causality is possible, particularly in the case of soy-protein formula which is often recommended for infants experiencing difficulties digesting dairy formula. It is possible that infants with a lower Z-scores in early infancy are more likely to be prescribed soy-protein formula.

It is interesting that these data do detect the previously reported observation of reduced WLZ among infants consuming partially hydrolyzed formula [12] among infants < 6 months of age. In this model, infants consuming fully hydrolyzed formulas showed a higher WLZ. However, only six infants < 6 months were consuming fully hydrolyzed formula in this cohort, and thus results may be spurious in this group.

Other relationships detected are thought provoking, including the positive relationship between amount of soy oil in formula and WLZ (in infants ≤ 12 months). This is clinically relevant as currently all US infant formulas in this dataset include soy oil in the fat blend at varying proportions.

A significant limitation of this dataset is the cross-sectional design. This does not allow for control for infants’ previous Z-scores (i.e.: where they are on the growth curve). Additionally, gestational age is not available in this dataset. All models did control for birth weight category which is a loose proxy for size at birth but does not remove the limitation. While no infants in this dataset were consuming premature infant formula, we are unable to control for infants born prematurely, but advanced enough in growth to be transitioned to standard formula. The 0–12 month age group contained both the <6 and 6–12 month group, and thus are not independent models. Additionally, these data are from the United States and popular infant formula ingredient sources may vary internationally. The strength of some of the relationships is modest which contributes to the limitations. Strengths include representation from a national population, a large sample size, and the novelty of the incorporation of formula macronutrient formulation. This is also a novel presentation of the macronutrient sources of formula fed infants’ diets.

In conclusion, these data provide evidence that the nutritional makeup of an infant’s formula needs to be included in considerations of growth outcomes in a clinical setting. Variation in ingredients within the standard formula category may indeed impact growth outcomes. Future research is needed to document the mechanistic effects of particular ingredients, and study long-term outcomes. Such studies are particularly needed for palm oil, sucrose, and soy protein.

## Figures and Tables

**Figure 1 nutrients-12-03465-f001:**
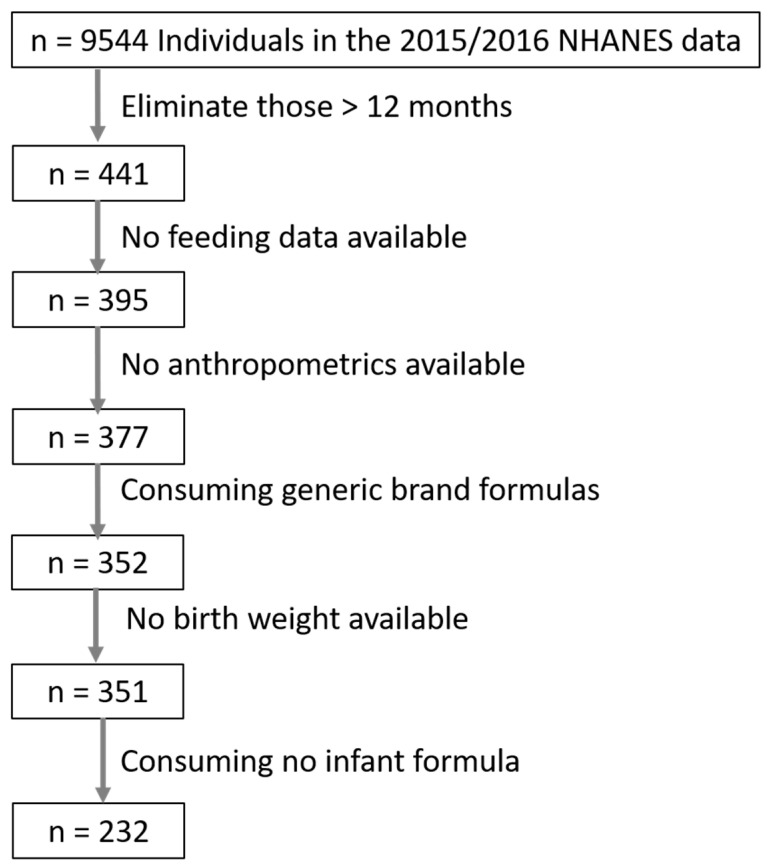
Sample sizes for generation of the final dataset of infants ≤ 12 months of age from the NHANES 2015–2016 dataset.

**Figure 2 nutrients-12-03465-f002:**
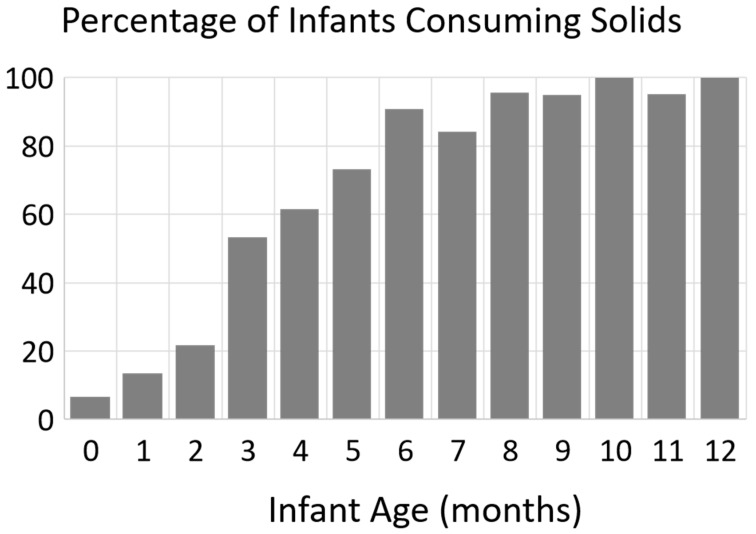
Presented are the percentage of infants that reported consuming any solid foods on any of the days when 24 h dietary recall was administered, by infant age in months.

**Table 1 nutrients-12-03465-t001:** Cohort Characteristics.

Characteristic	Mean ± SD or %	*n*
Infant Sex (% male)	47%	350
Birth Weight (g)	3278 ± 533	350
Birth Weight category	LBW = 7.7%; LGA = 6.6%	350
Breastfeeding Exclusivity ^1^ (all infants)	Exclusive Breastfeeding = 29% Mixed Feeding = 14% Exclusive Formula = 56%	325
Breastfeeding Exclusivity ^1^ (infants ≤ 6 months)	Exclusive Breastfeeding = 31% Mixed Feeding = 19% Exclusive Formula = 49%	180
Infant age at physical exam (months)	6.4 ± 3.6	350
Weight for age Z-score (WAZ)	0.47 ±1.05	323
Length for age Z-score (LAZ)	0.21 ± 1.1	323
Weight for Length Z-score (WLZ)	0.51 ± 0.99	323
Head Circumference for age Z-score (HCZ)	1.00 ± 1.22	196
Abdominal Circumference for age Z-score (ACZ)	0.47 ± 0.99	255

^1^ Based only on intake of breast milk and formula.

**Table 2 nutrients-12-03465-t002:** Average Infant formula macronutrient sources (*n* = 230).

Macronutrient Source	Mean ± SD or %	*n*
Daily caloric intake from formula (kcal)	531 ± 230 kcal	230
Protein size	Amino Acid Based = 0.4% Fully Hydrolyzed = 4.8% Partially Hydrolyzed = 22.4% Intact = 72.4%	228
Protein Source (if dairy formula)	Casein = 47.5 ± 25.0% Whey = 52.3 ± 25.2%	228
Protein source—% infants consuming soy	7.4%	228
Carbohydrate—Source	Lactose = 64.7 ± 41.4% Glucose = 31.7 ± 36.3% Sucrose = 3.4 ± 8.6%	228
Carbohydrate source—% infants consuming sucrose	15.4%	228
Fat Source	Palm Oil = 25.7 ± 22.4% Coconut/MCT Oil = 24.9 ± 5.9% Soy Oil = 25.4 ± 5.9%	228
Fat source—% infants by Palm Oil consumption categories	No palm oil = 41.2% 20–30% Palm oil = 3.1% ≥40% Palm oil = 55.6%	228

**Table 3 nutrients-12-03465-t003:** Model results for infant Z-scores among infants < 6 months and 6–12 months individually, and among all infants combined (0–12 months).

**A. Infants < 6 Months**		
**Independent Variable (Parameter)**	**Parameter** ***p*-Value**	**Parameter Estimate**
**WAZ (R^2^ = 0.10, *n* = 124)**		
Birth Weight category (reference = normal birth weight)	0.001	LBW: −1.04 ± 0.27, *p* = 0.0002 LGA: 0.923 ± 0.312, *p* = 0.003
**LAZ (R^2^ = 0.11, *n* = 124)**		
Birth Weight category (reference = normal birth weight)	0.001	LBW: −1.10 ± 0.29, *p* = 0.0003 LGA: 0.85 ± 0.33, *p* = 0.004
**WLZ (R^2^ = 0.08, *n* = 123)**		
Protein—Size (reference = intact protein)	0.087	Fully Hydrolyzed: 0.568 ± 0.267, *p* = 0.035 Partially Hydrolyzed: −0.245 ± 0.184, *p* = 0.185
Protein—Soy	0.143	Non-soy based: 0.265 ± 0.180
Fat—Palm Oil	0.076	Palm Oil-free: −0.185 ± 0.103, *p* = 0.076
Formula—kcal consumed/day	0.096	0.00062 ± 0.00037, *p* = 0.096
**HCZ (R^2^ = 0.03, *n* = 122)**		
Protein—Soy	0.048	Non-soy based: 0.424 ± 0.212
**ACZ (R^2^ = 0.10, *n* = 81)**		
Breastfeeding Category (reference = mixed fed)	0.032	Exclusively formula: 0.29 ± 0.13
Protein—Soy	0.059	Non-soy based: 0.43 ± 0.22
Fat—% Coconut oil	0.054	−1.108 ± 0.169 ^1^
**B. Infants 6–12 Months**		
**Independent Variable (Parameter)**	**Parameter** ***p*-Value**	**Parameter Estimate**
**WAZ (R^2^ = 0.15, *n* = 102)**		
Birth Weight category (reference = normal birth weight)	0.002	LBW: −1.00 ± 0.23, *p* < 0.0001 LGA: 0.96 ± 0.28, *p* = 0.0011
Fat—Palm Oil	0.264	Palm Oil-free: 0.11 ± 0.10
**LAZ (R^2^ = 0.17, *n* = 103)**		
Birth Weight category (reference = normal birth weight)	<0.0001	LBW: −1.03 ± 0.23, *p* < 0.0001 LGA: 1.12 ± 0.29, *p* = 0.0002
**WLZ (R^2^ = 0.06, *n* = 103)**		
Birth Weight category (reference = normal birth weight)	0.033	LBW: −0.634 ± 0.241, *p* = 0.0097 LGA: 0.534 ± 0.295, *p* = 0.073
Fat—Palm Oil	0.168	Palm Oil-free: 0.142 ± 0.102
**ACZ (R^2^ = 0.09, *n* = 102)**		
Carbohydrate—Sucrose	0.125	Sucrose-free: −0.182 ± 0.118
Birth Weight category (reference = normal birth weight)	0.008	LBW: −0.748 ± 0.235, *p* = 0.002 LGA: 0.752 ± 0.280, *p* = 0.008
**C. All Infants (0–12 Months)**		
**Independent Variable (Parameter)**	**Parameter** ***p*-Value**	**Parameter Estimate**
**WAZ (R^2^ = 0.13, *n* = 227)**		
Birth Weight Category (reference = normal birth weight)	<0.0001	LBW: −1.018 ± 0.175 (*p* < 0.0001) LGA: 0.925 ± 0.208 (*p* < 0.0001)
**LAZ (R^2^ = 0.13, *n* = 227)**		
Birth Weight Category (reference = normal birth weight)	<0.0001	LBW: −1.060 ± 0.184 (*p* < 0.0001); LGA: 0.986 ± 0.219 (*p* < 0.0001)
**WLZ (R^2^ = 0.031, *n* = 225)**		
Birth Weight Category (reference = normal birth weight)	0.029	LBW: −0.486 ± 0.181 (*p* = 0.0008); LGA: 0.471 ± 0.216 (*p* = 0.030)
Carbohydrate—Sucrose	0.073	Sucrose-free: −0.185 ± 0.103
Fat—% soy oil	0.122	0.570 ± 0.366 ^1^
**ACZ (R^2^ = 0.070, *n* = 183)**		
Birth Weight Category (reference = normal birth weight)	0.005	LBW: −0.655 ± 0.203 (*p* = 0.002) LGA: 0.685 ± 0.237 (*p* = 0.004)
Breastfeeding Category (reference = mixed fed)	0.043	Exclusively formula: 0.205 ± 0.100
Protein—Soy	0.205	Non-soy based: 0.187 ± 0.147
Carbohydrate—Sucrose	0.080	Sucrose-free: −0.175 ± 0.099

WAZ = Weight-for-age Z-score; LAZ = Length-for-age Z-score; WLZ = Weight-for-age Z-score; HCZ = Head circumference-for-age Z-score; ACZ = Abdominal circumference-for-age Z-score; LBW = Low Birth Weight; LGA = Large for Gestational Age. ^1^ For covariate variables that were transformed within models, parameter estimates reported were back-transformed in order to be clinically interpretable.

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
