# Peer review of "Variation in Infant Formula Macronutrient Ingredients Is Associated with Infant Anthropometrics"

_nutrients, 2020, doi:10.3390/nu12113465_

Round 1
Reviewer 1 Report
The question of infant growth based on variation in formula composition is a novel and interesting question, and I certainly appreciate the effort in tackling this issue. The first aim of demonstrating the variability of formula composition was achieved, although it should be specified that it describes the proportion of macronutrient intake of this specific patient cohort and not uniformly across multiple products. However, based on the data set used, it is not clear to me that we can truly arrive at any meaningful conclusions regarding infant growth.
As stated in the discussion, one of the limits to this study is its cross-sectional nature, but this is a major limitation, not a minor one. Because it only describes the single time point captured by the survey, it poses multiple problems. First, there are no additional time points from which to draw any trends. For example, what is the meaning of a z-score of 0.5 at 6 months of age if the birth z-score is unknown? The implications are different if the birth z-score was -0.5 vs 1.0, even though both of these would still fall under the “normal birth weight” category. Second, it is a largely variable time point as evidenced by the SD of 3.6 months, suggesting that this study design does not account for the heterogenous duration of exposure of each infant to those formulas. Lastly, as discussed by the author, the cross-sectional intake data may not adequately capture each infant's true intake from birth to survey.
In the discussion, there are also multiple attempts to draw a trend from 7 months to 12 months based on the 0-7 month subgroup analysis, but this cannot be done. First, the 0-12 month group includes the 0-7 month subgroup. It may be interesting to look exclusively at the 8-12 month infants as a separate subgroup. However, this does not solve the other issue that the 0-12 month (or 8-12 month) cohort does not actually provide a second time point for those same 0-7 infants, especially in light of how many variables do not actually overlap in the multivariate models between the two analyses.
That said, it may be more clinically relevant to report only on the infants whose intake is primarily formula or breast milk. This would allow for decreased heterogeneity in the sample population and reduce the confusion in group comparison. Furthermore, I would recommend reducing the cut off to less than or equal to 6 months to minimize caloric intake from solids, which may not be entirely insignificant in some infants as they border on 8 months.
Remainder of comments by line:
37: Please clarify if hydrolyzed protein comparison is relative to intact protein.
39: Consider adding background on whey and casein and why the proportion would have scientific plausibility in impacting growth (e.g. comparison to breast milk, digestibility/absorption).
79: Abdominal circumference is not a commonly utilized metric in practice to monitor growth or body composition in the pediatric population.
88-100: Specific patient exclusions should be reported in Results, not Methods.
103: Please clarify that “simple linear” refers to linear regression.
122: Table 1 – Is gestational age at birth available to add to this table? Would be helpful to know if these are all term infants or if there are any preterm or late preterm infants. Given that multiple points of the discussion focuses on calcium absorption and bone mineralization, the GA would be relevant, since preterm infants miss out on some degree of calcium/phosphorus accretion, which largely occurs in the third trimester.
131 & 152: Tables 3 & 4 - Is there a better way to organize this data? I found it difficult to follow, especially the non-reference categories in the third column. In addition, the model p-values were distracting from the parameter p-values -- might be cleaner to list/summarize the model p-values in the footnotes. Also, HCZ model is missing R2.
186: Can you comment on why you think your WAZ and LAZ models found a stronger significance and a more negative association with 20-30% palm oil intake and not >40%?
Reviewer 2 Report
Authors aimed at documenting the variation in macronutrient intake of formula fed infants in the United States and to investigate relationships between macronutrient composition and infant growth.
General critics:
This article is clear and original.
Point by point:
Methods:
lines 87-100: A flow chart would help the reader
lines 65-67 : it would be useful to cite (and classify) maltodextrin
line 81: HM should be defined
Results:
Table 2: I there a mistake in number of data in the protein size (higher than global sources) ?
Discussion:
The distribution of palm oil groups is imbalanced, which may weaken the relevance of results (as seen in table 3 and 4). This should at least be mentioned as a limitation
Line 235: The conclusion should begin with a new paragraph
Round 2
Reviewer 1 Report
Thank you for your edits, updated analysis, and answers to my questions. I appreciate the new division of subgroups of <6 mo and 6-12 and find that it adds clarity to your results.
My remaining comments are:
Lines 106-116 are still describing specific exclusions, though you moved the ones from the subsequent paragraph to the Results section as I had suggested previously. I'm not sure if this split was intentional; if so, it is actually more confusing now. I agree with the other reviewer than a flow diagram may actually be helpful for visualization and simplifying your text.
Lines 393 & onward: Recommend including in this discussion paragraph your response re: gestational age at birth being unavailable. My earlier comment was not in regards to palm oil exposures, but rather if there were any preterm infants, their bone mineralization at time of birth would be decreased compared to term counterparts, which could confound your conclusion re: bone growth. There is no standard approach or timing to transitioning preterm infants to term formula, so it's hard to know for sure based on the fact that none of them were receiving preterm formula at the time of the survey.
Author Response
Please see the attachment

This manuscript is a resubmission of an earlier submission. The following is a list of the peer review reports and author responses from that submission.